# Enhanced Polarization Properties of Holographic Storage Materials Based on RGO Size Effect

**DOI:** 10.3390/molecules29010214

**Published:** 2023-12-30

**Authors:** Jie Liu, Po Hu, Tian Ye, Jianan Li, Jinhong Li, Mingyong Chen, Zuoyu Zhang, Xiao Lin, Xiaodi Tan

**Affiliations:** 1College of Photonic and Electronic Engineering, Fujian Normal University, Fuzhou 350117, China; liujie_2120@163.com (J.L.); hupo_lzu_13@163.com (P.H.); tian.yeee@foxmail.com (T.Y.); ljn18816235361@163.com (J.L.); ljh0401_fjnu@163.com (J.L.); mingychen@foxmail.com (M.C.); fjunnin9@163.com (Z.Z.); 2Key Laboratory of Opto-Electronic Science and for Medicine of Ministry of Education, Fuzhou 350117, China; 3Fujian Provincial Key Laboratory of Photonics Technology, Fuzhou 350117, China; 4Fujian Provincial Engineering Technology Research Center of Photoelectric Sensing Application, Fuzhou 350117, China

**Keywords:** reduced graphene oxide, size effect, photopolymer, orthogonal polarization holography

## Abstract

Polarized holographic properties play an important role in the holographic data storage of traditional organic recording materials. In this study, reduced graphene oxide (RGO) was introduced into a phenanthraquinone-doped polymethylmethacrylate (PQ/PMMA) photopolymer to effectively improve the orthogonal polarization holographic properties of the material. Importantly, the lateral size of RGO nanosheets has an important influence on the polymerization of MMA monomers. To some extent, a larger RGO diameter is more conducive to promoting the polymerization of MMA monomers and can induce more PMMA polymers to be grafted on its surface, thus obtaining a higher PMMA molecular weight. However, too large of a RGO will lead to too much grafting of the PMMA chain to shorten the length of a single PMMA chain, which will lead to the degradation of PQ/PMMA holographic performance. Compared with the original PQ/PMMA, the diffraction efficiency of the RGO-doped PQ/PMMA photopolymer can reach more than 11.4% (more than 3.5 times higher than the original PQ/PMMA), and its photosensitivity is significantly improved by 4.6 times. This study successfully synthesized RGO-doped PQ/PMMA high-performance photopolymer functional materials for multi-dimensional holographic storage by introducing RGO nanoparticles. Furthermore, the polarization holographic properties of PQ/PMMA photopolymer materials can be further accurately improved to a new level.

## 1. Introduction

Volume holographic storage technology stores data in a three-dimensional space using light waves as the information carrier [1,2,3]. Holographic storage material is the medium of holographic storage technology [4,5], which needs to meet high sensitivity, fast response speed, simple preparation, and low-cost performance. The advantages of photopolymers [6], such as high optical sensitivity, low manufacturing cost, infinite size, and unnecessary post-processing, are important materials for holographic applications, such as data storage [7] and holographic optical components [8]. The phenanthraquinone-doped polymethylmethacrylate (PQ/PMMA) photopolymer is one of the best candidates for holographic storage materials due to its low raw material cost, simple preparation process, negligible shrinkage, controllable thickness, and polarization recording ability [9,10]. However, the low-polarization holographic optical properties, such as the diffraction efficiency and photosensitivity of PQ/PMMA materials when recording polarization information, also limit the large-scale application in holographic data storage [11,12]. The low polarization sensitivity is mainly due to the low saturated solubility and intermolecular diffusivity of photosensitizer PQ in methyl methacrylate (MMA) [13]. Various methods have been proposed to improve the saturated solubility of PQ and the holographic properties of PQ/PMMA materials. Some studies have tried to improve the holographic properties of PQ/PMMA materials by doping other materials such as nanoparticles, organometallic components, and comonomers [6,14,15,16]. However, the ideal effect is not obtained in improving the orthogonal polarization holographic performance of PQ/PMMA. Different from traditional intensity holography, orthogonal polarization holography uses two orthogonal polarized lights to perform interference recording, and the intensity distribution formed by the interference regions of the two orthogonal polarization lights is uniform. There is no interference region of intensity modulation. Introducing polarization modulation can enhance the dimensionality of holographic storage. The holographic storage density can be improved by increasing the orthogonal polarization enhancement of the polarization sensitivity of the material. Considering the low orthogonal polarization holographic properties of PQ/PMMA photopolymer materials, our group grafted a polymethyl methacrylate (PMMA) macromolecular polymer by introducing reduced graphene oxide (RGO) [17,18,19].

Reduced graphene oxide (RGO) has an ultra-high elastic modulus and strength [20,21], excellent thermal and electrical conductivity [22], huge specific surface area [23], and optical properties [24,25], and is widely used in composite materials [26,27], fuel cells [28], electrocatalysis [29], optoelectronic devices [30,31], and other field watchers. Moreover, RGO is a nanosheet particle with a single-layer sheet structure, and its surface and edges of the RGO obtained via reduction contain a small amount of functional groups, such as unsaturated carbon–carbon double bonds, and a small amount of oxygen-containing functional groups such as hydroxyl and carboxyl groups on the basal surface, making it more hydrophilic and dispersive. Integration with other polymer materials can be achieved by dispersing RGO in aqueous solutions or some organic solutions [32,33,34,35], such as doping it as a comonomer of the polymer, which not only exhibits miscibility with the polymer matrix but also the useful properties of polymers can be obtained, to obtain new high-molecular polymer hybrid materials required for a wide range of applications [36,37,38].

In this experiment, the functional groups and end groups on the RGO structure were used to induce the grafting of the PMMA polymer onto the RGO nanosheet substrate (Figure 1). It was found that the holographic optical properties of the synthesized RGO-PMMA/PQ photopolymer composites were significantly improved. The N-methylpyridinone (NMP) solvent was used as the RGO protective liquid in the experiment. It was mentioned in previous work that NMP was only an intermediate solvent, did not react with monomers, and had excellent solubility to RGO [10,16]. Furthermore, the relevant properties of RGO are strongly dependent on its lateral dimension [39,40,41,42], which provides the possibility of improving the holographic optical properties of photopolymers.

We report a method for precise size grading based on the diffusion distribution of RGO nanosheets in aqueous solvent dispersions, which can be the pristine RGO nanosheets that are graded into arbitrarily different ideal sizes with a narrow size distribution, and further analyze the effect of the RGO size effect on enhancing the holographic performance of PQ/PMMA photopolymer materials. In addition, to further analyze the chemical behavior of the internal structure of the material, various microscopic characterization methods were used to deeply analyze the interaction mechanism between RGO and PQ/PMMA polymer materials. The study found that, on the one hand, RGO can promote the polymerization of MMA to form the macromolecular PMMA polymer and then induce the grafting behavior of PMMA with it. On the other hand, the size effect of RGO plays an important role in regulating the molecular weight of the macromolecular polymer PMMA, and the mechanical strength and mechanical properties of RGO of different sizes are different [39,43,44]; this has a significant effect on small molecular monomers (including MMA and PQ) that also produce different degrees of physical adsorption. Combined with the influence of the size effect of RGO nanosheets, this study further analyzes the microscopic mechanism of RGO in PQ/PMMA photopolymers, which provides a new strategy for regulating the macroscopic material properties from the microscopic level.

## 2. Results

### 2.1. Holographic Characteristics

This study uses the experimental optical system of orthogonal polarization holography [45] with an s polarization wave as a signal beam and a p polarization wave as a reference beam, as shown in Figure 2. Both the signal beam and the reference beam used a 532 nm green laser with a light intensity of 0.102 W cm^−2^. In the recording stage process, the interference angle between the signal beam and the reference beam was 24°, the recording was performed for 6 s, and the reconstruction time was set to 0.4 s for reproduction with the same reference beam.

The two processes of recording–reproducing are repeated until the photosensitive molecules are completely depleted at this location. The diffraction efficiency *η* is defined as [46]:(1)$$\eta =\frac{I_{+1}}{I_{0}+I_{+1}}$$
where *I*_0_ is the intensity of transmitted light, and *I*_+1_ is the intensity of first-order diffracted light. RGO-PMMA/PQ photopolymer materials were prepared by setting different concentrations of RGO, and the RGO doping mass concentration gradients were 0.7, 1.0, 1.3, 1.6, and 2.0 × 10^−3^ wt%, respectively.

Compared with the pristine PQ/PMMA photopolymer, the orthogonal polarization diffraction efficiency of the RGO-PMMA/PQ photopolymer is significantly improved. When the RGO concentration is 1.3 × 10^−3^ wt%, the orthogonal diffraction efficiency is relatively best, and its diffraction efficiency reaches about 8.0%. As the RGO doping concentration continues to increase, the diffraction efficiency will gradually decrease, as shown in Figure 3. To further evaluate the effect of the RGO size effect on the holographic optical properties of PQ/PMMA photopolymers, we prepared RGO nanosheets of different sizes, with sizes of 5 μm, 10 μm, 15 μm, 20 μm, 25 μm, and 30 μm, respectively, as shown in Figure 4, and the doping concentration was set to 1.3 × 10^−3^ wt%. In each experiment, pure PQ/PMMA and original RGO were set as control groups for the convenience of subsequent analysis and comparison. At this concentration, the orthogonal polarization diffraction efficiency of RGO-PMMA/PQ photopolymers prepared via doping RGO with different sizes was tested, and the diffraction efficiency gradually increased with the increase in RGO size. Moreover, when 20 μm size RGO nanosheets were used for doping, the diffraction efficiency of the RGO-PMMA/PQ photopolymer was as high as 11.4%, but as the size of RGO continued to increase, the diffraction efficiency showed a downward trend. The orthogonal polarization diffraction efficiency of the RGO-PMMA/PQ photopolymer is more than 3.5 times higher than that of the pristine PQ/PMMA photopolymer, as shown in Figure 5a. The pristine data curve represents the PQ/PMMA material without RGO added. In addition, compared with the pristine PQ/PMMA photopolymer material, the introduction of RGO significantly enhanced the photosensitivity of the PQ/PMMA photopolymer material, as shown in Figure 5b. The photosensitive coefficient *S* of the material is expressed using the diffraction efficiency, which is defined as [15]:(2)$$S=\frac{1}{Id}(\frac{\partial \sqrt{\eta }}{\partial t})$$
where *I* is the recording signal light intensity (0.102 W/cm^2^), *d* is the material thickness (1.5 mm), and *η* is the grating diffraction efficiency. The test results show that the photosensitivity is improved from 0.9 × 10^−2^ cm/J to 4.2 × 10^−2^ cm/J when doped with 20 μm of RGO, which is 4.6 times higher than the original PQ/PMMA photopolymer material.

On the other hand, a photo-induced birefringence optical system was set up to test the photo-induced anisotropy of the material, as shown in Figure 6. The red light with a wavelength of 632.8 nm is used as the probe light, and the green light with a wavelength of 532 nm is used as the pump light. Furthermore, the probe light is incident on the surface of the material through a negative 45° polarizer P1, with a light power of 0.6 mW and a beam diameter of 2 mm. After penetrating the material, the probe light continues to pass through a positive 45° polarizer P2 and finally hits the photodetector. The pump light first passes through an attenuator to control the power of the light, then passes through a beam expander to control the beam diameter, then passes through a polarizer P0 to irradiate the photo-induced birefringence of the material surface, and finally hits the photodetector. In order to reduce the experimental error, the pump light needs to be incident on the surface of the material approximately parallel to the probe light, and the included angle between the two beams is 6°. The power and beam diameter of the pump light is set to 20 mW and 5 mm, respectively.

The photo-induced birefringence of the RGO-PMMA/PQ photopolymer was tested to determine the ability of the sample-polarized holographic recording, which can directly reflect the storage density, storage capacity, and polarization sensitivity of the polarized holographic material. The addition of RGO can effectively improve the refractive index modulation of the PQ/PMMA photopolymer, as shown in Figure 7, and the change trend is highly consistent with the diffraction efficiency. The refractive index modulation is an important indicator that can reflect the storage density and capacity of the material. According to the Kogelnik coupled-wave theory [47], the refractive index modulation Δn of the material is related to the diffraction efficiency *η* of the grating as follows:(3)$$\Delta n(t)=\frac{\lambda \cos\theta_{0}}{\pi d}\arcsin\sqrt{\eta }$$
where *λ* is the recording light wavelength (532 nm), *η* is the diffraction efficiency of the grating, *θ*_0_ is the angle between the recording signal light and the recording reference light, *d* is the thickness of the material, and t is the holographic recording time.

The above experimental results show that the introduction of RGO nanosheets into the PQ/PMMA photopolymer can effectively improve the polarization holographic performance of the material. Different sizes of RGO can improve the holographic optical performance of the photopolymer differently; the polarization holographic properties of PQ/PMMA photopolymers increased first and then decreased with the increase in RGO size. RGO doping with a size of 20 μm had the best effect on improving the polarization holographic properties of PQ/PMMA photopolymers. It is proved that the size effect of RGO significantly improves the holographic optical properties of PQ/PMMA photopolymer materials. The prepared holographic materials with excellent polarization properties can be used to improve the holographic properties of vortex beams and further obtain holographic optical devices with excellent properties [48].

### 2.2. Stability Study of RGO-PMMA/PQ Photopolymer

In this study, the holographic optical properties of the material were analyzed by combining microscopic characterization and macroscopic testing. To further verify the stability and physical and mechanical properties of the cured material, the volume shrinkage of the RGO-PMMA/PQ photopolymer material was tested. Since volume shrinkage can seriously affect the practical application of photopolymers in holographic performance, the volume shrinkage of RGO-PMMA/PQ materials was evaluated in this study using the Bragg diffraction angle shift. The angle multiplexing experiment was used in this experiment [49,50], and the main lobe Bragg angle was measured using a computer-controlled Sigma (OSM-60YAW, Sigma, Tokyo, Japan) motorized rotary stage; set the resolution per pulse to 0.005° and the rotation speed to 0.25°/s. The volume shrinkage of the RGO-PMMA/PQ photopolymer material is shown in Figure 8. The volumetric shrinkage coefficient *σ* is defined as [51]:(4)$$\sigma =1-\frac{\tan\theta_{t}}{\tan\theta_{e}}$$
where *θ*_t_ is the theoretical value of the Bragg angle after interference, and *θ*_e_ is the experimental value of the Bragg angle after interference. The samples were made of photopolymer materials with a thickness of 0.5 mm, and a computer-controlled electric rotating stage was used. The test results show that the Bragg angle selectivity bandwidth of the PQ/PMMA material is about 0.05°, and the peak shift of the RGO-PMMA/PQ material is about 0.01°, with a symmetrical and sharp normal curve distribution. According to Equation (4), the volume shrinkage *σ* of the polymer material was calculated. Compared with PQ/PMMA, the *σ* of the RGO-PMMA/PQ material was reduced from 0.5% to 0.1%, and the inhibition range was further reduced, indicating that the volume shrinkage in the PQ/PMMA photopolymer material by introducing RGO is negligible.

## 3. Reaction Mechanism

In order to analyze the effect of RGO on the material deeply, various microscopic characterization methods were used to test and analyze the polymer material. The RGO-PMMA polymer was prepared using free-radical polymerization [52,53,54].

The Fourier-transform infrared (FTIR) spectra of PMMA, RGO, and RGO-PMMA polymers were tested, as shown in Figure 9a. The FTIR spectra clearly indicated that RGO might induce the grafting reaction of the PMMA polymer, and the characteristic peaks of pristine RGO, PMMA, and RGO-PMMA polymer were compared. The original RGO has characteristic absorption peaks at 3437 cm^−1^ (O-H), 2916 cm^−1^ (C-H), 1725 cm^−1^ (C=O), 1569 cm^−1^ (C=C), and 1211 cm^−1^ (C-C) [55], and observed that the C=O bond of RGO at 1725 cm^−1^ increases sharply after thermal polymerization, the characteristic absorption peak of C=C bond at 1569 cm^−1^ decreases, and there is a slight red shift (~9 cm^−1^), the characteristic absorption peak of C-C bond at 1211 cm^−1^ has a significant red shift to 1267 cm^−1^, which may be due to the thermal polymerization of RGO and PMMA is completed by consuming the C=C bond, and caused by the C=O bond contained in PMMA. In addition, RGO showed increased O-H bonds at 3437 cm^−1^ and C-H bonds at 2916 cm^−1^, which were covered by the characteristic absorption peaks of PMMA at 1070 cm^−1^ (C-O-C) and 1441 cm^−1^ (C-H). The above results indicate that a strong interaction between RGO and PMMA has been formed, and the reaction site is on the C=C bond. Raman spectroscopy (RAMAN) was also used to characterize RGO and RGO-PMMA composites, as shown in Figure 9b. The D peak is a defect peak, reflecting the existence of sp^3^-disordered hybrid carbon in the RGO structure, and the G peak reflects its symmetry and crystallinity. The D peak detected via RGO and RGO-PMMA appeared at 1345 cm^−1^, and the G peak appeared near the position of 1593 cm^−1^. At this time, compared with the intensity ratio (I_D_/I_G_) of D peak and G peak before and after thermal polymerization of RGO and RGO-PMMA (I_D_/I_G_ = 1.07) was significantly increased compared with RGO (I_D_/I_G_ = 0.94), indicating that the crystal defects of carbon atoms increased, that is, some carbon atoms are converted from sp^2^ hybridization to sp^3^ hybridization, which may be attributed to the reduction of the average size of sp^2^ domains in RGO caused via PMMA during surface treatment.

The mass loss changes of PMMA, RGO, and RGO-PMMA were analyzed using thermogravimetric analysis (TGA) curves, as shown in Figure 10a. The RGO-PMMA polymer material has two stages of mass loss, which appear near the temperature positions of 75 °C and 290 °C, respectively. The two stages correspond to the mass loss of RGO and PMMA under their respective temperature conditions. The first mass loss should be caused by the thermal decomposition of unstable functional groups on the RGO basal surface, and the second mass loss stage is due to the degradation of the PMMA polymer. These experimental phenomena show that PMMA causes the mass change of the RGO-PMMA polymer, which proves the graft modification behavior induced by the PMMA polymer. Furthermore, to further analyze the various elemental compositions and molecular structures of the polymer materials, the X-ray photoelectron spectroscopy (XPS) maps of RGO and RGO-PMMA polymers were tested, as shown in Figure 10b. RGO and RGO-PMMA polymers have two obvious electron-binding energy peaks: the C peak at 284 eV and the O peak at 533 eV, respectively. The ratio of a carbon atom and an oxygen atom concentration of RGO and RGO-PMMA polymers (C/O) were 94:6 and 87:13, respectively, and the carbon-to-oxygen ratio of RGO-PMMA decreased significantly, which was mainly due to the formation of more C=O and O-C-O after the polymerization of RGO and PMMA, as shown in Figure 11. Further, it is indicated that a strong interaction is formed between RGO and PMMA, which is consistent with the previous FTIR test results.

To further analyze the microstructure and morphology of the polymer materials, the pristine RGO and RGO-PMMA polymers were dispersed in absolute ethanol, respectively, and characterized via transmission electron microscopy (TEM, JEOL, Tokyo, Japan) and high-resolution transmission electron microscopy (HRTEM, JEOL, Tokyo, Japan). RGO itself is a single-layer nanosheet structure, showing a transparent single-layer sheet-like structure under electron microscopy, as shown in Figure 10c. After grafting PMMA, it can be observed that the RGO-PMMA polymer exhibits enhanced wrinkles and roughness, as shown in Figure 10d, and the transparency of the RGO-PMMA polymer was significantly reduced, and its surface showed an inhomogeneous state. These results demonstrate the successful grafting of PMMA onto the RGO surface.

In addition, atomic force microscopy (AFM) was used to analyze the changes in the surface morphology and height of the samples. An AFM model SPM-9700 was used to test the surface and structure of the sample using a dab mode. The sample was dispersed with an ethanol solution and spun onto a freshly peeled mica substrate for testing. The thickness of graphene is about 0.3 nm. The surface of RGO obtained via reduction contains a small amount of oxygen-containing functional groups. The thickness of RGO is about 0.7 nm, as shown in Figure 12a, which is consistent with the material parameters provided by the manufacturer. The thickness of RGO-PMMA was detected to be about 3 nm, as shown in Figure 12b, and its cross-sectional image that the basal plane of the RGO-PMMA sheet is more blurred than that of the RGO sheet. AFM images of different sizes of RGO and RGO-PMMA polymers are shown in Figure 13. The large thickness variation trend also indicated that the RGO surface was grafted with PMMA macropolymers rather than small monomers.

To avoid the light-absorption characteristics brought via RGO itself, the UV–Vis absorption spectra of the PQ/PMMA photopolymer and RGO-PMMA/PQ photopolymer prepared via RGO of different sizes were tested, as shown in Figure 14a. The absorption spectra of RGO-PMMA/PQ photopolymers doped with different sizes have a good overlap with those of PQ/PMMA photopolymers, which clearly shows that the introduction of RGO does not affect the PQ/PMMA photopolymers spectral absorption. Furthermore, there is a lower absorption near the wavelength of 532 nm, and the light absorption is almost zero near the wavelength of 632.8 nm. Therefore, the green laser at 532 nm can be used as the pump source, and the red laser at 632.8 nm is suitable as the detection source. Considering that different sizes of RGO have an important effect on the holographic optical properties of PQ/PMMA photopolymers, gel permeation chromatography (GPC) was used to test the changes of the PMMA molecular weight caused by different sizes of RGO doping and GPC elution, the curve is shown in Figure 14b. Compared with the pristine PMMA polymer, the introduction of RGO significantly increased the weight-average molecular weight (M_w_) of PMMA, and with the increase in the size of the doped RGO, the M_w_ gradually increased, and the PDI decreased significantly. When the RGO size was 20 μm, the M_w_ and PDI reached the relative optimum values of 272.190 kg/mol (unit of M_w_) and 1.089, respectively. Continuing to increase the RGO size, the M_w_ showed a downward trend, further showing that the RGO size effect affects the generation and grafting behavior of PMMA. This is in good agreement with the variation of the orthogonal polarization diffraction efficiency of the material tests. This is due to the strong physical adsorption of RGO itself [56], and the larger the size of the RGO sheet, the stronger its adsorption capacity. The decrease in molecular weight may be due to the strong interaction between the oversized RGO, which causes competition in the polymerization of MMA, resulting in the shortening of the PMMA chains. Likewise, the large-diameter RGO would be more competitive for the attractiveness of PQ photosensitizers, resulting in the photopolymers exhibiting lower holographic optical properties.

## 4. Material Preparation

### 4.1. Preparation of RGO with Different Lateral Dimensions

In the experiment, we set up a set of engineering techniques to size-classify RGO nanosheets, as shown in Figure 15, using ethanol as the dispersing solvent for RGO nanosheets. Teflon membranes with uniform pore size were used for filtration in a filtration device to fractionate RGO nanosheets into fractions of different sizes. Theoretically, RGO nanosheets with smaller lateral dimensions can pass through the membrane pores, while relatively large RGO nanosheets can block the membrane pores. To prevent the RGO nanosheets from clogging the filter membrane pores, during the filtration process, a stirring paddle was prepared to stir the solution [57], and the vacuum pump was used to accelerate the filtration and adjust the power to appropriate power. In this way, the RGO nanosheets can be quickly and successfully classified into two parts: the RGO nanosheets smaller than the filter membrane in the filtered filtrate, and the RGO nanosheets larger than the pore size of the filter membrane are left on the filter membrane. In addition, after the RGO nanosheets on the filter membrane were removed from the surface of the filter membrane, no residual RGO nanosheets were found on the filter membrane, the filter element, and the surface of the filter, indicating that the filter is durable and can be reused for RGO nanosheet particle size classification. Remove the RGO nanosheets on the filter membrane, redissolve them, and select different-size filter membranes according to actual needs to obtain RGO nanosheets of any size, and then use a centrifuge and set an appropriate number of revolutions to centrifuge the RGO nanosheet dissolving solution, the larger size RGO nanosheets will sink to the bottom of the centrifuge tube in the form of a precipitate, and the small size RGO nanosheets will appear in the upper part of the centrifuge tube in the form of a dispersion liquid, and then take the sediment at the bottom of the centrifuge tube again for redissolving [32,33,58], and perform a new round of filter filtration and centrifuge centrifugation. The overall operation of filtration-centrifugation was repeated about three times to accurately obtain RGO nanosheets with the desired lateral dimensions. The RGO nanosheets in each remaining filtrate can be used for the next size fractionation to obtain RGO nanosheets of different sizes.

### 4.2. Preparation of RGO-PMMA Polymers

The RGO-PMMA polymer was prepared using free-radical polymerization, and the preparation process is shown in Figure 16. First, the RGO nanosheets were dispersed in N-methylpyridinone (NMP) to obtain a well-dispersed RGO suspension supernatant, and then the monomer MMA and thermal initiator AIBN were added in sequence. The mixture was stirred in a 333 K constant temperature water bath at 660 r/min for 75 min. After stirring, the mixture was poured into methanol to wash away small molecular monomers and impurities to obtain macromolecular polymer precipitates. The precipitates were taken out and dried in a 313 K oven, and then the dried precipitates were dissolved with NMP and transferred to a centrifuge tube, centrifuged at 12,000 r/min for 1 h with a centrifuge, take the sediment at the bottom of the centrifuge tube and repeat the centrifugation operation until the supernatant of the centrifuge tube is colorless. Then, the precipitate was dissolved with acetone instead of NMP and centrifuged about three times. Finally, the polymer precipitate was transferred to a 313 K oven for drying for 24 h to obtain the pure RGO-PMMA polymer.

### 4.3. Preparation of the RGO-PMMA/PQ Photopolymer

In order to prepare the RGO-PMMA/PQ photopolymer material, the MMA monomer was first added to the 3 mL reaction flask. The photosensitizer PQ (1.3 wt%) and the thermal initiator AIBN (1 wt%) were added to the reaction flask and mixed uniformly, and then the pre-prepared RGO solution was added to make it evenly mixed. The reaction flask containing the uniformly mixed solution was placed in a magnetic stirrer with a constant temperature water bath of 333 K, and the stirring was continued at a speed of 660 r/min for about 1 h until the solution became viscous. Then, the viscous liquid was poured into a special glass mold with a thickness of 1.5 mm, clamped and sealed with clips, and placed in a 333 K oven for thermal polymerization for about 20 h until the material was cured, the RGO-PMMA/PQ photopolymer material with high-quality optical properties can be obtained [59], as shown in Figure 17.

This experiment combines two processes of stirring pre-polymerization and thermal polymerization, which can effectively promote the polymerization of MMA monomers, and the formed PMMA substrate effectively avoids photo-induced shrinkage during photo-polymerization.

## 5. Conclusions

In summary, we successfully synthesized the RGO-PMMA/PQ photopolymer using radical polymerization and set different concentration gradients of RGO nanosheets, which proved that the introduction of RGO nanosheets could effectively improve the PQ/PMMA photopolymer and the orthogonal polarization holographic performance. At the same time, we developed a low-cost, high-practical value, and scalable engineering technique to achieve the precise size grading of RGO nanosheets. This technique can obtain RGO nanosheets of any desired size with a narrow size distribution. Larger-sized RGO nanosheets exhibit denser, more ordered, and less-defective microstructures. In addition, the size effect of RGO nanosheets plays an important role in improving the holographic properties of PQ/PMMA photopolymers. With the increase in the size of the RGO nanosheets, the holographic properties of the PQ/PMMA photopolymer showed a trend of first increasing and then decreasing. Compared with the pristine PQ/PMMA photopolymer, the orthogonal diffraction efficiency of the RGO-PMMA/PQ photopolymer prepared via doping with 20 μm RGO nanosheets can be significantly improved to about 11.4%. On the other hand, using various microscopic characterization methods (such as FTIR, Raman, XPS) demonstrated that RGO nanosheets could promote the polymerization of monomeric MMA to form PMMA polymer and induce the grafting of the PMMA polymer to the RGO basal plane through carbon–carbon double bonds, the holographic optical properties can be improved by adjusting the molecular weight of the PMMA polymer. In this study, the holographic optical properties of PQ/PMMA photopolymers were improved by introducing RGO nanosheets, and the size effect of RGO nanosheets brought more excellent properties to PQ/PMMA photopolymers. More importantly, the RGO-PMMA/PQ photopolymer material has the characteristics of long-term preservation and negligible volume shrinkage and has great potential for holographic storage applications. This study uses the size effect of RGO nanosheets to tune the holographic optical properties of PQ/PMMA photopolymers, which provides a new strategy for regulating the properties of macroscopic materials at the microscopic level.

## Figures and Tables

**Figure 1 molecules-29-00214-f001:**
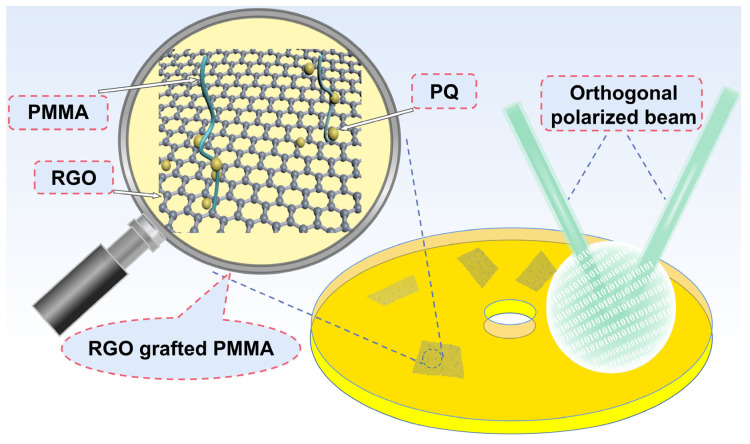
Schematic diagram of RGO-PMMA/PQ formed using free-radical polymerization.

**Figure 2 molecules-29-00214-f002:**
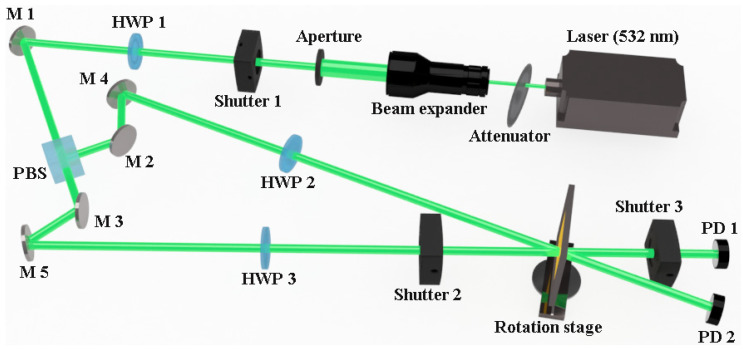
Experimental device for measuring diffraction efficiency of holographic recording, in which HWP: half-wave plate; PBS: polarizing beam splitter; and PD: photo detector.

**Figure 3 molecules-29-00214-f003:**
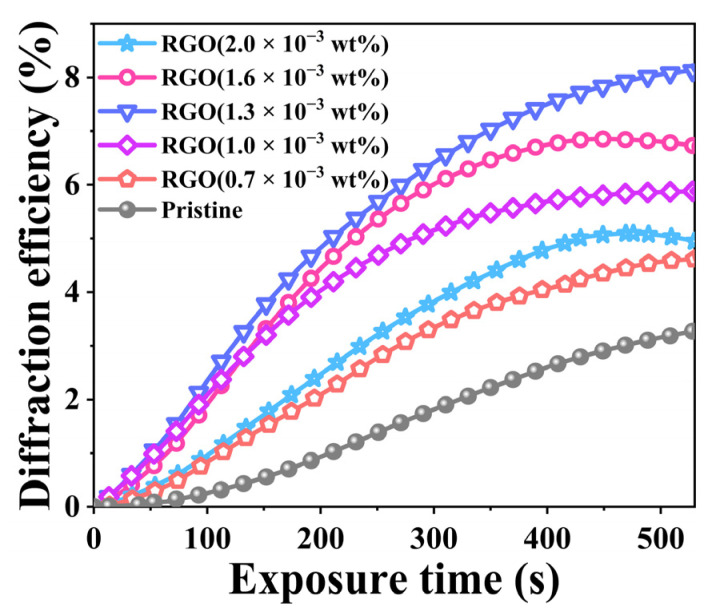
Time-dependent polarization holographic diffraction efficiency of pristine PQ/PMMA and different concentrations of RGO-PMMA/PQ polymers.

**Figure 4 molecules-29-00214-f004:**
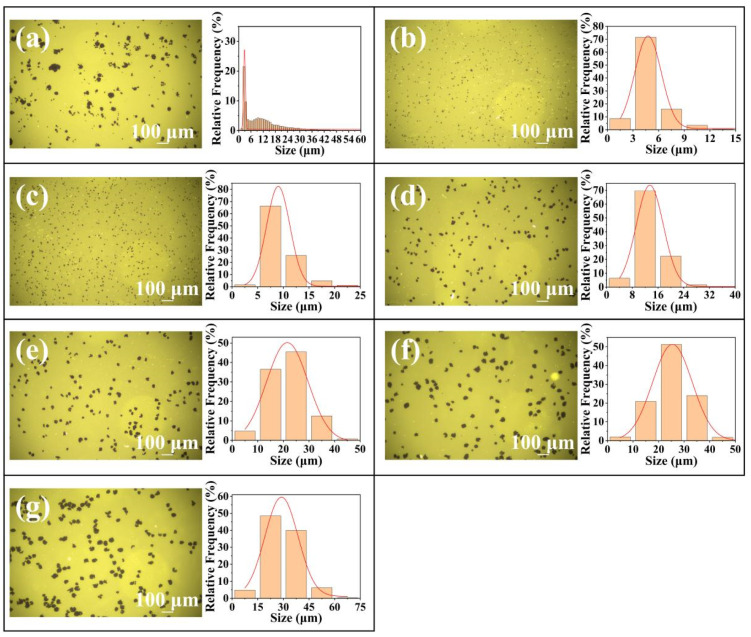
Size statistical results of RGO and (**a**) not graded, (**b**) 5 μm, (**c**) 10 μm, (**d**) 15 μm, (**e**) 20 μm, (**f**) 25 μm, and (**g**) 30 μm. The red curve indicates the Gaussian fitting change of each sample.

**Figure 5 molecules-29-00214-f005:**
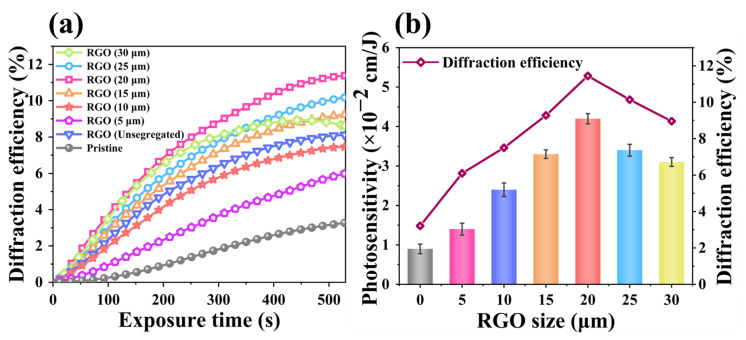
(**a**) Time-dependent polarization holographic diffraction efficiencies of pristine PQ/PMMA and RGO-PMMA/PQ polymers of different sizes. (**b**) Photosensitivity and diffraction efficiency of RGO-PQ/PMMA polymers of different sizes.

**Figure 6 molecules-29-00214-f006:**
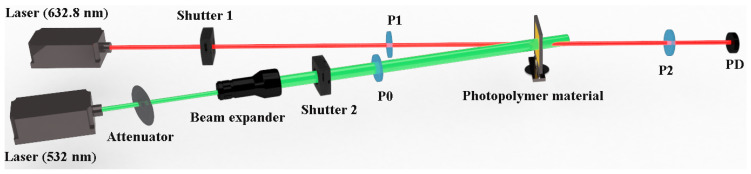
Photo-induced birefringence optical path diagram, P0: horizontal polarizer, P1: negative 45° polarizer, and P2: positive 45° polarizer.

**Figure 7 molecules-29-00214-f007:**
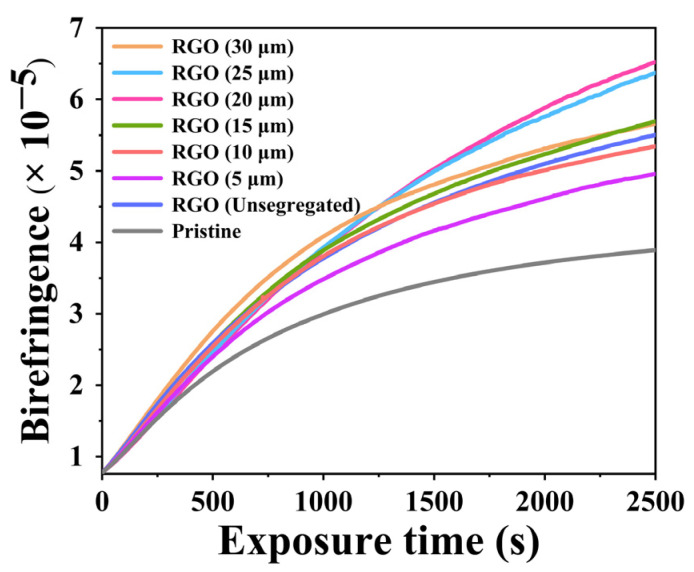
Birefringence modulation of exposure intensity of different sizes of RGO-doped PQ/PMMA and original PQ/PMMA with time.

**Figure 8 molecules-29-00214-f008:**
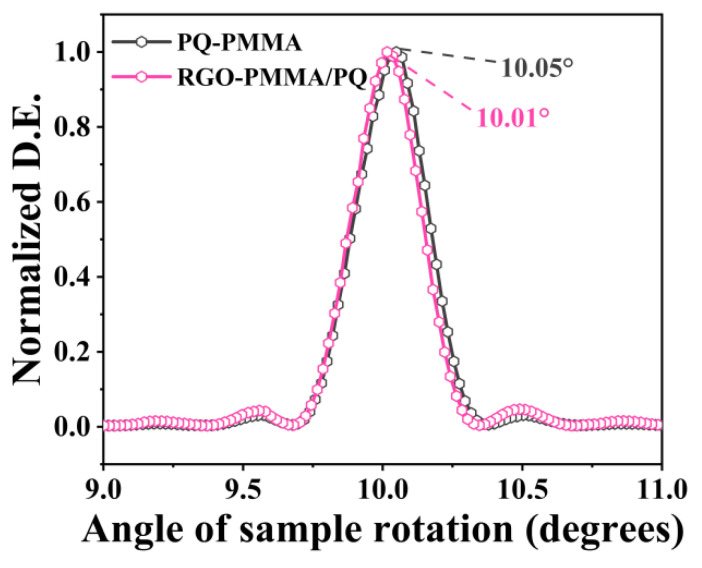
Normalized diffraction efficiencies of 0.5 mm thick RGO-PMMA/PQ and PQ/PMMA samples set to rotate 10° from the bisector of the two incident beams as a function of the sample rotation angle.

**Figure 9 molecules-29-00214-f009:**
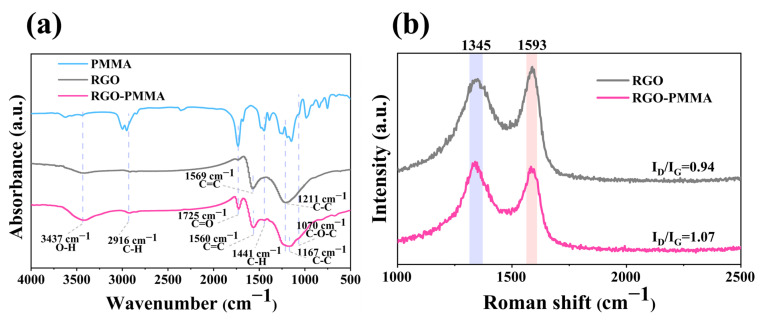
(**a**) FTIR spectra of RGO, PMMA, and RGO-PMMA. (**b**) Raman spectra of RGO and RGO-PMMA.

**Figure 10 molecules-29-00214-f010:**
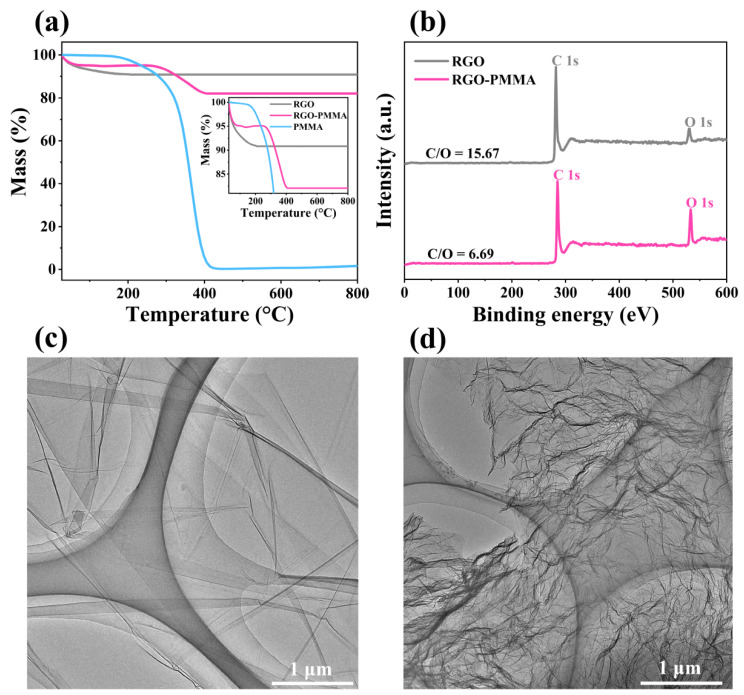
(**a**) Thermogravimetric analysis spectra of RGO, PMMA, and RGO-PMMA. (**b**) XPS spectra of RGO and RGO-PMMA. TEM images of (**c**) original RGO and (**d**) RGO-PMMA with a resolution of 1 μm.

**Figure 11 molecules-29-00214-f011:**
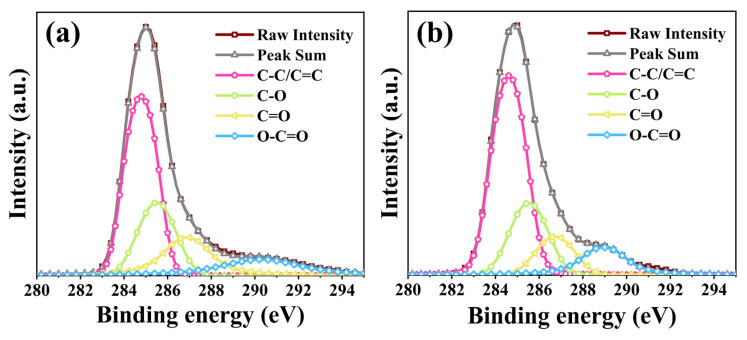
C1s XPS spectra of (**a**) original RGO and (**b**) RGO-PMMA polymer.

**Figure 12 molecules-29-00214-f012:**
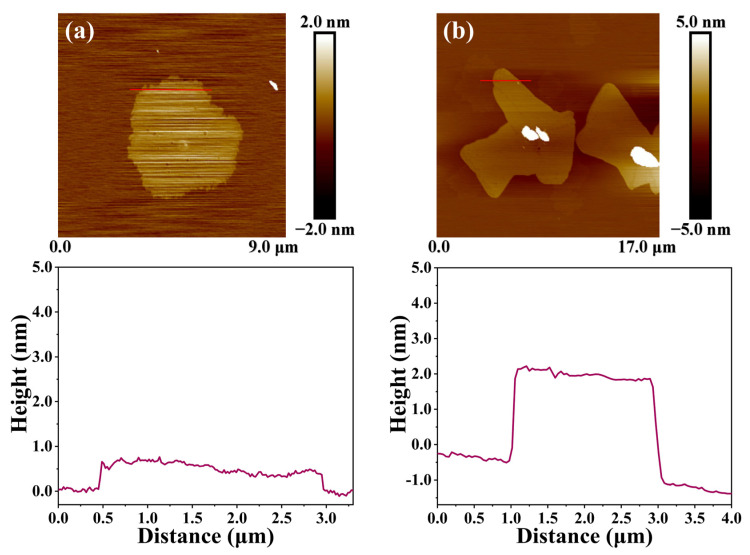
AFM images of (**a**) RGO and (**b**) RGO-PMMA.

**Figure 13 molecules-29-00214-f013:**
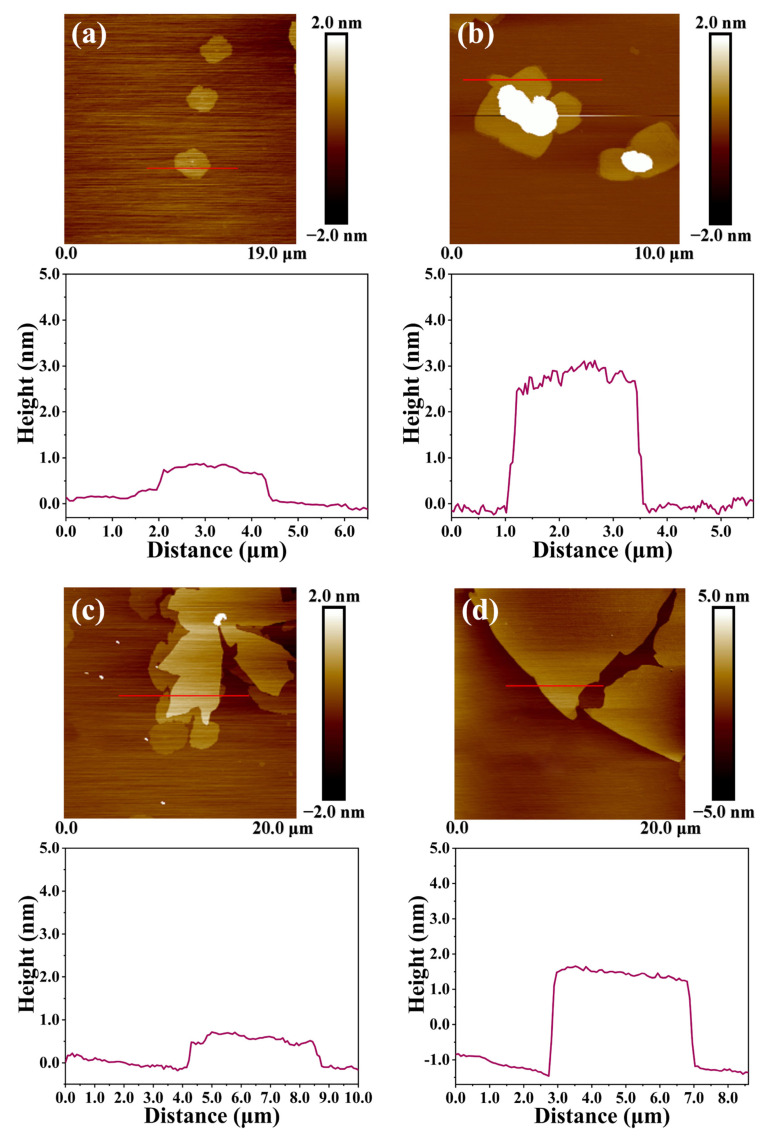
AFM images and thickness change diagrams of (**a**,**c**) original RGO and (**b**,**d**) RGO-PMMA polymers with small and large sizes, respectively.

**Figure 14 molecules-29-00214-f014:**
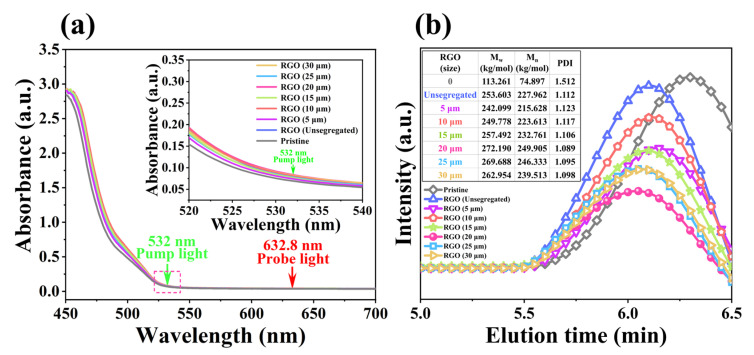
(**a**) UV–Vis absorption spectra of pristine PQ/PMMA and RGO-PMMA/PQ of different sizes. The inset is the UV–Vis absorption spectrum for wavelengths between 520 nm and 540 nm. (**b**) GPC elution curves, weight average molecular weight (M_w_), number average molecular weight (M_n_), and polydispersity index (PDI) of RGO-PMMA of different sizes.

**Figure 15 molecules-29-00214-f015:**
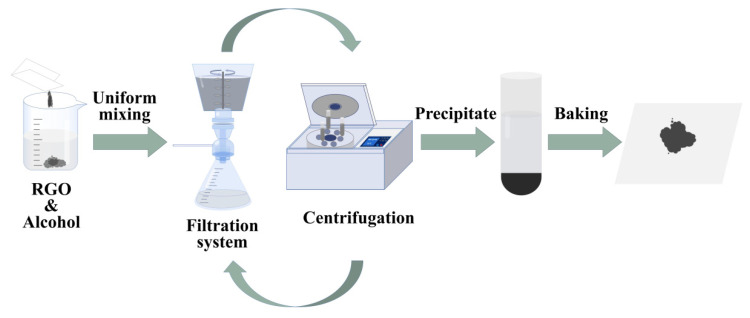
Schematic diagram of RGO size-grading process.

**Figure 16 molecules-29-00214-f016:**
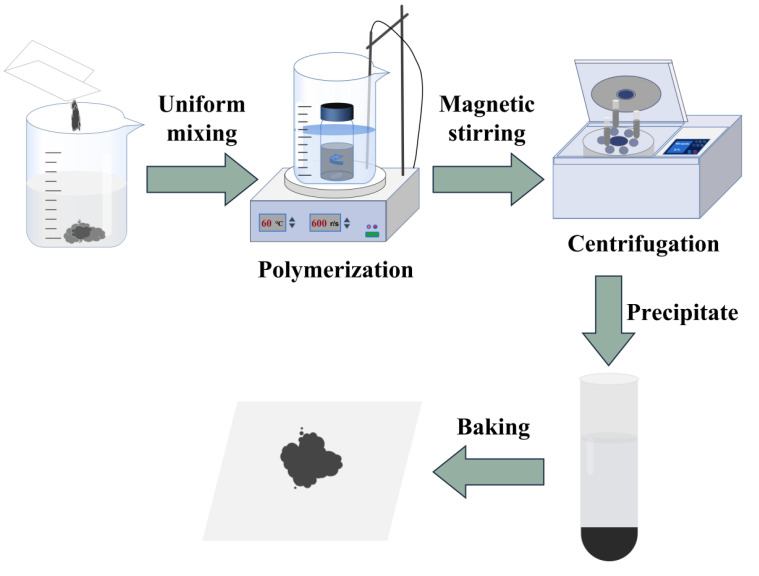
Schematic diagram of the purification process of RGO-PMMA.

**Figure 17 molecules-29-00214-f017:**
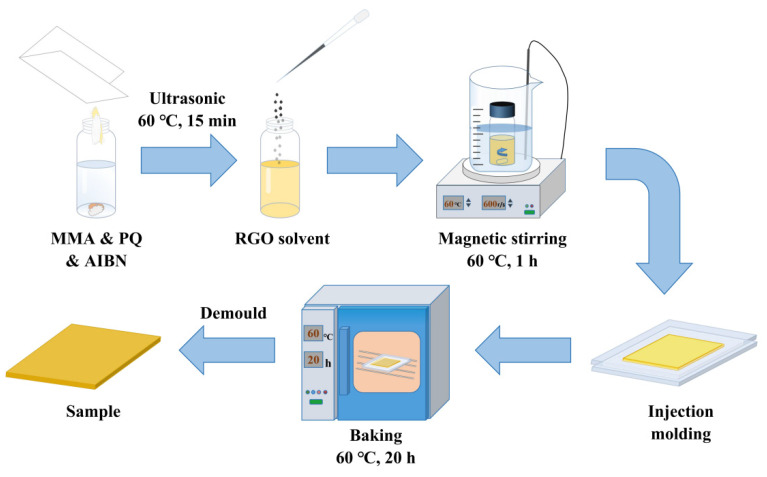
Schematic of preparation process of RGO-PMMA/PQ photopolymers.

## Data Availability

Data are contained within the article.

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
