# Peer review of "Enhanced Polarization Properties of Holographic Storage Materials Based on RGO Size Effect"

_molecules, 2023, doi:10.3390/molecules29010214_

Round 1
Reviewer 1 Report
Comments and Suggestions for Authors
The research article is good written and work is promising. However some areas of the paper may need more attention. The introduction part lack the explicit vision and significance of work. In your article, most references are 5 or more years old. This will be better to include latest references.
Author Response
We thank the reviewers for their careful reading of our manuscript. The reviewers realize that we have done interesting research work in the synthesis of high-sensitivity holographic polymers, which may have a significant impact on the data storage application of holographic materials. As the reviewers pointed out, articles containing the latest references are very important and necessary for our manuscript, and the introduction of these articles will make our research more attractive and attractive. We added some latest references to the revised manuscript.
Reviewer 2 Report
Comments and Suggestions for Authors
The authors reduced graphene oxide (RGO) was intro-12 duced into phenanthraquinone doped polymethylmethacrylate (PQ/PMMA) photopolymer to ef-13 fectively improve the orthogonal polarization holographic properties of the material. Importantly, 14 the lateral size of RGO nanosheets has an important influence on the polymerization of MMA mon-15 omers. The material synthesis and characterization are well organized. The novelty of the work is enough to publish in Molecules. However, I have a few minor comments on the manuscript. The authors have to address my concerns before accepting it to publish in Molecules.
1. Authors have to recheck the technical part of the manuscript. Especially in sections 2 and 3.
2. Please cite the references for the equations used in this manuscript. If the authors derived them for the first time in this manuscript, then they have to provide more details about them.
3. The quality of Figure 2 has to be improved.
4. More information about figures should be provided in the captions of Fig. 2, 3, and 10.
5. The inset of Fig. 16 is not clear.
6. The experimental setup used in Fig. 4 and its results are similar to the previous work of Rao, A.S., Dar, M.H., Venkatramaiah, N., Venkatesan, R. and Sharan, A., 2016. Third order optical nonlinear studies and its use to estimate thickness of sandwiched films of tetra-phenyl porphyrin derivatives. Journal of Nonlinear Optical Physics & Materials, 25(03), p.1650039. Is there any correlation between them?
7. Authors can cite previous similar kinds of works in section 3.1 so that readers can understand easily.
8. Why authors have used orthogonally polarized beams, especially s and p polarizations? Is there any necessity?
9. In Fig. 5, why is the diffraction efficiency saturating with references to exposure time and is it linear or nonlinear? Also, some of them decrease with exposure time after a certain time around 450 seconds.
10. Authors can provide details of AFM used in the text.
11. Authors should provide more details on the possible applications of their work. It will be very interesting and useful for the readers.
Comments on the Quality of English LanguageThe quality of the English writing in the technical part must be improved.
Author Response
About the response, please check the attachment. Thank you so much.

Reviewer 3 Report
Comments and Suggestions for Authors
The manuscript did not emphasize why size effects can enhance its holographic polarization performance. The properties of RGO were described in detail in the paper, but how does its size effect play a role in holographic diffraction performance? Please provide further detailed explanation. And what are the differences and advantages of doping RGO compared to GO in its polarization performance?
(1)Line 54: “The orthogonal polarization the enhancement of the polarization sensitivity of the material is achieved by increasing the holographic storage density.” How to understand that increasing the holographic storage density can improve the polarization sensitivity of materials??
(2)Line 78-81: In previous work, it was mentioned that NMP was only used as an intermediate solvent and did not participate in the reaction with the monomer, and was not suitable for the reaction between the monomers [13,21], and excellent solubility for RGO. “did not participate in the reaction with the monomer, and was not suitable for the reaction between the monomers”
How to understand the sentence? What is the meaning of “not suitable for the reaction between the monomers”, it may be repeated with the previous one “participate in the reaction with the monomer”
(3)Line 330-332:“And there is a lower absorption near the wave number of 532 nm, and the light absorption is almost zero near the wave number of 632.8 nm.”
Here, 532 nm is not corresponding to wavenumber but to wavelength.
Why the authors select 532 nm with a lower absorption as pump source region?
It is rather low absorption, which is similar to 671 nm, what’s the difference between them?
(4)In conclusion, “the RGO-PMMA/PQ photopolymer material has the characteristics of long-term preservation”
How you confirm the long life of RGO-PMMA/PQ, any supporting information?
(5)Figure 6, size distribution of RGO, how to obtain it, please describe in detail.
(6)Line 135: “recording was performed for 6 s, and the reconstruction time was set to 0.4 s for reproduction with the same reference beam.” please explain the difference when comparing Exposure time in Figure 5.
(7)Line 160: “pure PQ/PMMA and original RGO”, is it corresponding to RGO (Unsegregated) in Figure 7(a)?
(8)Line 170, Figure 7(b), Add arrows or text to illustrate dotted and bar charts
(9)Line 210, “θ0 is the angle between the recording signal light and the recording reference light”
2θ0 is the angle between the recording light and the reference light.
(10)Line 338: “weight average molecular weight (Mw)”, W should be in the form of a subscript.
“number average molecular weight (Mn)”is not mentioned.
(11) “4. Stability Study of RGO-PMMA/PQ Photopolymer” , RGO size?
The thickness of the film is also different from what was mentioned earlier.
Comments on the Quality of English Language
Please check the manuscript in detail, there are many errors in unit abbreviation and sentence form, such as um,FT-IR,RAMAN,Photo induced birefringence et al.
Author Response

(The authors gave the same response as above.)

Round 2
Reviewer 3 Report
Comments and Suggestions for Authors
The current version can be published.
Comments on the Quality of English LanguageThe current version can be published.